# Moderate Protein Restriction in Advanced CKD: A Feasible Option in An Elderly, High-Comorbidity Population. A Stepwise Multiple-Choice System Approach

**DOI:** 10.3390/nu11010036

**Published:** 2018-12-24

**Authors:** Antioco Fois, Antoine Chatrenet, Emanuela Cataldo, Francoise Lippi, Ana Kaniassi, Jerome Vigreux, Ludivine Froger, Elena Mongilardi, Irene Capizzi, Marilisa Biolcati, Elisabetta Versino, Giorgina Barbara Piccoli

**Affiliations:** 1Néphrologie, Centre Hospitalier Le Mans, 72000 Le Mans, France; antiocofois@gmail.com (A.F.); antoine.chatrenet@gmail.com (A.C.); emanuela.cataldo@gmail.com (E.C.); flippi@ch-lemans.fr (F.L.); akkaniassi@ch-lemans.fr (A.K.); jevigreux@ch-lemans.fr (J.V.); lfroger@ch-lemans.fr (L.F.); 2SCDU Urology, Department of Oncology, ASOU San Luigi, University of Torino, 10043 Orbassano, Italy; elenamongilardi@yahoo.it; 3Nephrology, Department of Clinical and Biological Sciences, ASOU San Luigi, University of Torino, 10043 Orbassano, Italy; irene.ccapizzi@gmail.com; 4Obstetrics, Department of Surgery, Città della Salute e della Scienza, University of Torino, Torino 10126, Italy; marilisa.biolcati@unito.it; 5SS Epidemiology, Department of Clinical and Biological Sciences, ASOU San Luigi, University of Torino, 10043 Orbassano, Italy; elisabetta.versino@unito.it

**Keywords:** chronic kidney disease, protein restriction, protein intake, obesity, diabetes, compliance

## Abstract

Background: Protein restriction may retard the need for renal replacement therapy; compliance is considered a barrier, especially in elderly patients. Methods: A feasibility study was conducted in a newly organized unit for advanced kidney disease; three diet options were offered: normalization of protein intake (0.8 g/kg/day of protein); moderate protein restriction (0.6 g/kg/day of protein) with a “traditional” mixed protein diet or with a “plant-based” diet supplemented with ketoacids. Patients with protein energy wasting (PEW), short life expectancy or who refused were excluded. Compliance was estimated by Maroni-Mitch formula and food diary. Results: In November 2017–July 2018, 131 patients started the program: median age 74 years (min–max 24-101), Charlson Index (CCI): 8 (min-max: 2–14); eGFR 24 mL/min (4–68); 50.4% were diabetic, BMI was ≥ 30 kg/m^2^ in 40.4%. Normalization was the first step in 75 patients (57%, age 78 (24–101), CCI 8 (2–12), eGFR 24 mL/min (8–68)); moderately protein-restricted traditional diets were chosen by 24 (18%, age 74 (44–91), CCI 8 (4–14), eGFR 22 mL/min (5–40)), plant-based diets by 22 (17%, age 70 (34–89), CCI 6.5 (2–12), eGFR 15 mL/min (5–46)) (*p* < 0.001). Protein restriction was not undertaken in 10 patients with short life expectancy. In patients with ≥ 3 months of follow-up, median reduction of protein intake was from 1.2 to 0.8 g/kg/day (*p* < 0.001); nutritional parameters remained stable; albumin increased from 3.5 to 3.6 g/dL (*p* = 0.037); good compliance was found in 74%, regardless of diets. Over 1067 patient-months of follow-up, 9 patients died (CCI 10 (6–12)), 7 started dialysis (5 incremental). Conclusion: Protein restriction is feasible by an individualized, stepwise approach in an overall elderly, high-comorbidity population with a baseline high-protein diet and is compatible with stable nutritional status.

## 1. Introduction

End-stage renal disease (ESRD) is a condition in which high mortality and poor quality of life combine with high costs [1,2,3]. In high-income countries, the incidence of patients starting renal replacement therapy is stable or increasing and about one individual out of 1000 is treated by dialysis or transplantation [1].

Elderly and high-comorbidity patients are the most relevant and most rapidly increasing subset of the uremic population; in this fragile subset the advantages of correction of uremia may be offset by the hyatrogenicity of dialysis, thus leading to an emergent trend of developing alternative approaches with supportive measures [4,5,6]. 

While data are conflicting and difficult to combine, the available evidence suggests that survival is overall higher in elderly CKD patients managed by dialysis, but their quality of life may be worse (despite much higher costs) [4,5,6,7,8,9]. Mortality is extremely high, especially in the first months after dialysis starts, where, according to the DOPPS data, it may reach up to 33 deaths per 100 patient-years, and this finding should induce us to focus on two main issues: how to safely prolong the dialysis-free interval, and how to smooth the transition to dialysis [9,10,11].

Protein restriction is considered a valuable tool for retarding the need for starting renal replacement therapy [12,13,14]. Conversely, high protein intake is believed to contribute to preserving nutritional status in elderly patients or in patients with high comorbidity [15,16,17,18]. How to mediate between these conflicting requirements in elderly or high-comorbidity CKD patients is not clear, as very few studies have addressed these subsets of the population [19,20].

Furthermore, because of low adherence to dietary recommendations, doctors often decide not to prescribe them, in particular for patients in which the cause of renal failure is linked to dietary habits, as is often the case in obese patients or those with type 2 diabetes [21,22,23,24]. In particular, in the elderly, or diabetic and obese patients, baseline dietary habits are considered to be extremely difficult to change, thus reducing the acceptance of protein restriction. Likewise, Mediterranean countries, such as Italy, where low-protein diets are a part of the routine management of CKD patients, are usually thought to be favored by baseline patterns, unlike other areas of Europe, including central and northern France, where the present study was performed [25,26,27]. In these settings, as well as in the United States or Australia, the generally higher protein intake in fact makes it difficult to convince patients of the need to follow protein-restricted diets [28,29,30,31,32]. 

However, even though the literature often classifies protein intake in terms of three “magic numbers”, i.e., normal protein intake (0.8 g/kg/day), moderate protein restriction (0.6 g/kg/day), and very low protein intake (0.3 g/kg/day), the effect of protein restriction is more likely to be a continuum and some studies demonstrated that each reduction of at least 0.2 g/kg/day has a protective effect on kidney function and can retard the need for renal replacement therapy [29,30,31]. Data from countries, such as the United States or Australia, in which protein intake is overall very high, confirm these findings and suggest that even normalization of protein intake may be a reasonable, nutritionally safe goal [28,29,30,31,32].

Previous experience with a multiple-choice diet system, developed in a northern Italian setting, in which Mediterranean dietary patterns are widespread, showed that an approach offering different options, at various levels of protein restriction, produced favorable results in elderly and diabetic patients [33,34]. 

Along with this experience, to which we will refer for comparison, the present study tests the hypothesis that personalization of nutritional approach represents a feasible option for first normalizing and then reducing protein intake in a CKD population with high median age and comorbidity, followed up in Le Mans, France, a region where baseline protein intake is high and dietary habits are more meat-based than plant-based. 

## 2. Materials and Methods

### 2.1. Study Setting; Patient Selection and Inclusion Criteria

The study was carried out in a newly founded unit for the care of advanced kidney disease (called UIRAV, from Unité pour l’Insuffisance Rénale chronique AVancée), which opened on 15th November 2017 in the Centre Hospitalier Le Mans (CHM), in central France. The CHM is one of the largest non-university hospitals in France, with about 1750 beds (16 in the nephrology ward), offering care to patients from the initial stage of kidney disease, to dialysis and follow-up after transplantation (which is performed by neighboring university hospitals). The unit’s patients include individuals with advanced CKD or with particular clinical or psychological needs.

CKD was defined according to the usual definitions of the K/DOQI clinical practice guidelines; e-GFR was assessed by the CKD-EPI equation. 

While all patients are assessed for dietary habits, and reduction of protein intake is advised in cases in which protein intake exceeds 1 g/kg/day, in keeping with the indications of the World Health Organization, patients with CKD stages 3–5 (who are not dialysis dependent) start a conjoint nephrology and nutrition follow-up, whose initial results are discussed in this paper. No protein restriction is prescribed to patients who refuse to comply, and conversely, dietary supplementation is prescribed in the case of PEW, observed mainly in patients with neoplasia, short life expectancy or multiple, severe comorbidity. 

All patients are followed up to identify signs of protein energy wasting (PEW), such as reduction in body weight (unexplained by edema reduction), reduction in lean body mass (clinical assessment, and integration with bioimpedance on demand), reduction in serum albumin, prealbumin or total proteins, especially in the absence of acute inflammatory event, or other clinical marker of poor nutrition, from the dietary journal, assessed by the dietitians, in response to vitamin deficits or unexplained anemia.

### 2.2. Diets and Controls

In this setting, a nephrologist and/or a dietitian assess baseline protein intake, and the nephrologist establishes a first prescription of normalization or reduction in protein intake based on a series of parameters including baseline protein intake, nutritional status, trajectory of CKD progression, proteinuria, age, comorbidity and life expectancy. The clinical suggestions (normalization of protein intake versus reduction of protein intake) are extensively discussed with the patient, and the main nutritional strategy (mixed proteins or plant based) is chosen (Figure 1).

Furthermore, since in most cases the baseline dietary protein intake is high, a stepwise approach, from normalization to restriction, is established and progressively pursued (Figure 2). The patients are followed-up by the nephrology and dietetics group throughout the whole process of protein reduction; moreover, the patients who do not undergo a protein restricted diet are also followed-up to timely identify and try to compensate for the eventual signs of malnutrition. This multiple-choice stepwise approach represents an adaptation to French dietary patterns of a “diet system” previously set up in Italy, a country in which it is possible to rely on widespread acceptance of a Mediterranean baseline pattern, and reduction of protein intake is facilitated by the availability of protein-free commercial food, provided free of charge to CKD patients (Figure 3) [25,33,34].

“Traditional” diets are based upon the analysis of the usual dietary patterns in the area (for example, they rely on one mainly vegetarian meal per day, based upon a vegetable soup, containing potatoes as a source of starch, and small portions of dairy products, according to the usual habits in the French countryside), and maintain that about 50% of the protein intake should be of animal origin. Conversely, plant-based diets rely on carbohydrates such as potatoes, rice, bread, pasta as main sources of calories, and privilege proteins of vegetable origin (from grains and beans); since these protein sources are usually incomplete as for essential amino acids, supplementation with a mixture of amino acids and ketoacids (Kestosteril, available free of charge for CKD patients in both Italy and France) is added. The dose, for moderately restricted diets, in keeping with the previous Italian experiences, is 1 tablet per 8–10 kg of body weight, to be further adjusted on the basis of albumin levels or protein losses [25,33,34]. Very low protein diets are always integrated by ketoacids and amino acids (Ketosteril, 1 tablet per 5 kg of real body weight).

In the case of obese patients, the prescription of protein is based on actual weight up to a BMI of 35 kg/m^2^ and to adjusted body weight above this limit.

Energy intake is aimed at 30–35 kcal/kg/day in non-obese younger individuals and at a minimum of 25 kcal/kg/day in elderly patients, provided that their weight and nutritional status remain stable. Energy intake is assessed by the dietitian using the patient’s 3- to 7-day food diary or, in its absence (non-adherence, older age, etc.) based on the patient’s dietary recall. 

Energy intake is established on a case-by-case basis in obese patients, on the account not only of caloric intake, but also of the daily activities, privileging, wherever possible, increasing physical activity to reducing energy below 25 Kcal of adjusted weight per day. 

Sodium intake is normalized where needed, with a first goal of less than 6 g of NaCl per day, which is slightly higher than the recommendation of the World health organization (2 g sodium/day, equivalent to 5 g salt/day), but is in line with a policy of progressive normalization of all crucial intakes, to favor compliance. Due to the wide use of diuretics in our population, further adjustments are done, when needed, on the basis of the sodium level, of the overall diet and of hydration level. 

Supplementation with calcium, sodium bicarbonate, vitamin D, folic acid, vitamin B12, iron and erythropoietin is tailored to blood levels on the basis of the usual indications of clinical best practice. Biochemical tests and routine visits are scheduled every 1–2 months in stable subjects and up to once a week for patients with severe metabolic derangements or when GFR drops to below 7–10 mL/min. Dialysis start is decided in CKD stage 5 on the basis of the clinical picture (assessment includes anorexia, weight loss, nausea, malnutrition, restless leg syndrome, poorly controlled hypertension); GFR, urea levels, water and acid-base balance, calcium-phosphate-PTH balance, anemia and serum albumin are taken into account in the evaluation of the patient. 

### 2.3. Biochemical Data, Compliance and Nutritional and Comorbidity Indexes

Biochemical parameters are assessed in the patient’s laboratory of choice, and in CHM’s general laboratory during periodic hospitalization in “day hospital” (Outpatient clinic dedicated to treatments that have to be done in-hospital, but do not need overnight stay, such as iron infusion, or IV diuretics, or to complex diagnostic assessments, including at least three consultations or imageries). The following data are regularly checked: serum creatinine (normal range: 0.6–1.2 mg/dL), BUN (6–21 mg/dL), eGFR (CKD-EPI equation), proteinuria (0.0–0.3 g/day); HCO3 (23–31 mmol/L), albumin (3.5–5.0 g/dL), parathyroid hormone (PTH, 13–46 ng/mL), vitamin D (30–80 µg/L), calcium (2.1–2.55 mmol/L). 

For the sake of uniformity in dosage methods, only the results of tests performed in the hospital’s general laboratory were included in the analysis of compliance. 

Compliance with the protein intake is assessed whenever possible with the Maroni-Mitch formula, and with the concomitant analysis of a food diary, assessed by the renal-diet team, four dietitians trained in the care of CKD and dialysis patients [35]. When 24-h urine collection is not feasible (incontinence, logistic problems in very old patients) or not done (dementia, logistic problems, non-compliance), the analysis is based on a food diary or dietary recall (with patient or caregiver). In the case of a discrepancy between the two measures, the case is discussed with the senior nephrologist, with direct experience in dietary prescription and management, and the most reliable measure is retained. 

The following indexes were assessed by the nephrologist for all patients at the start of follow-up in UIRAV, with a planned update after one year of observation: Charlson Comorbidity Index (CCI) (scale: 0–33); subjective global assessment (SGA, A, B or C); Malnutrition inflammation score (MIS, scale: 0–30) [36,37]. All data was reviewed by the senior nephrologist. 

Compliance with the prescribed diet was assessed separately for the two main levels of protein restriction (0.8 g/kg/day and 0.6 g/kg/day). For the logistic analysis, only patients with at least 3 months of follow-up were included.

Good compliance was defined as protein intake equal to or below prescription; a tolerance interval of 20% was accepted for “0.6 diets” and of 15% for “0.8 diets”. The definition was based on the Maroni-Mitch formula when available, or using the patient’s food diary, assessed by an experienced dietitian, in the other cases. 

## 3. Statistical Analysis

A descriptive analysis was performed as appropriate: median and [min-max] for non- parametric data, mean and standard deviation for normal distribution. The Shapiro-Wilk Test was used to assess normality of distribution. 

Statistical significance was assessed by ANOVA for parametric data, Kruskal-Wallis for non-parametric data, in compliance with standard indications for continuous variables. Dichotomous data are presented as risks, rates and proportions; in this case the significance of the differences was tested using the Chi-Square Test.

The significance of pre-post differences was tested using the T-Test for normally distributed variables and the Wilcoxon Signed Rank Test for variables that were not normally distributed. The McNemar Test was used to compare pre-post qualitative data, and statistical significance was set as >0.05 for all tests.

### 3.1. Logistic Regression

Two sets of logistic regression analyses were performed: in the first, “good compliance” was considered as the dependent variable (good compliance was defined as prescribed protein intake plus a maximum of 20%, meaning max 0.96 g/kg/day and 0.72 g/kg/day respectively for the 0.8 g/kg/day and 0.6 g/kg/day diets). The following explicative variables were tested: sex, BMI (dichotomized at 30 kg/m^2^), eGFR, dichotomized at <20 mL/min. To avoid collinearity, the CCI and diabetes were entered in separate models, since the CCI includes diabetes. Likewise, since age is included in the CCI, age was considered in the model including diabetes (and not in the CCI) and two separate analyses (with and without age) were performed for the model that included the CCI (collinearity between age and comorbidity is incomplete). 

In the second model, the outcome “good diet”, defined as following a 0.6 g/kg/day diet with good compliance; the same explicative covariates were tested: sex, BMI, eGFR; the CCI and diabetes, as described above. 

Analysis was performed using SAS System 9.4 (SAS Institute Inc., North Carolina, NC, USA) 

### 3.2. Ethical Issues

The study was conducted in accordance with the Declaration of Helsinki.

The Italian reference study was approved by the ethics Committee of the University of Torino, Italy (PROTERENE, delibera 282, 28/01/2015). 

The protocol of the French study was approved by the Ethics Committee of the Centre Hospitalier Le Mans (avis favourable, séance du 14/06/2018).

Informed consent was obtained for anonymous management of clinical data from each patient at the start of follow-up in each Center. Further consent for publication was not needed for this study, dealing with overall data, not with individual cases. 

## 4. Results

### 4.1. Baseline Data

The study population consists of 131 patients who were evaluated and/or followed up in the UIRAV, the Unit dedicated to patients with advanced CKD, which started regular activity on 15th November 2017 (Figure 1). 

The population followed so far is relatively old (median age 74 (min-max: 24–101), 49 patients (37.4%) are more than 80 years old (age distribution is depicted in Figure 4)), and with high comorbidity (median on the Charlson Index – CCI- is 8 (min-max: 2–14), with 22 patients (16.8%) with an index of 10 or above; 66 patients (50.4%) are diabetics). These data may explain some of the differences observed in the diet distribution and main results compared with an Italian population that was previously followed adopting a similar approach (Table 1, Figure 3). In fact, the French cohort is characterized by significantly older age, higher comorbidity, higher prevalence of diabetes (roughly half of the patients) and of nephroangiosclerosis, in spite of a similar degree of kidney function impairment at recruitment (Table 1). 

The distribution of the kidney function at recruitment (CKD-EPI) is depicted in Figure 5; of note, eGFR was above 45 mL/min in 4 cases, who were started on protein restriction on the account of a combination of rapid decrease of the kidney function, nephrotic proteinuria or single kidney. 

### 4.2. Choice of Diet

Table 2 reports the main characteristics of the population, sorted according to the diet prescribed at most recent update (31st July 2018 for patients continuing follow-up). 

Only 10 patients, with high comorbidity, were not prescribed a diet, given their very complex clinical situation, and short life expectancy; in fact, 5 of them died before the end of the observation period (Table 2 and Table 3).

Otherwise, the main clinical characteristics of the patients who chose different diets are significantly different: patients for whom normalization was chosen are overall older and at higher comorbidity, while younger patients tended to choose plant-based supplemented diets. Furthermore, supplementation was added to 0.8 diets in 10 patients presenting signs of protein-energy wasting, such as low body weight, low albumin level, or nephrotic syndrome. 

Interestingly, in the overall population, protein energy malnutrition and low BMI are exceedingly rare, and about 40% of the population is obese (Table 4).

### 4.3. Special Populations: Obese Patients

In this elderly, high comorbidity population, obesity is prevalent (median BMI 28.3kg/m^2^ [16.7–51.2]; 53 patients (40.4%) have BMI at or above 30 kg/m^2^, 25 (19%) at or above 35 kg/m^2^ and 10 patients (7.6%) at or above 40 kg/m^2^). Besides BMI, and the prevalence of diabetes, which are higher in the obese cohort, no functional or demographic characteristic distinguishes the population of obese and non-obese patients. 

Interestingly, in the descriptive analysis, compliance with protein restriction is higher, albeit non-significantly, for obese patients: considering 86 patients with at least 2 controls, compliance was rated as good in 85% of the obese versus 75% of the non-obese patients. Indeed, these patients significantly decrease their protein intake, and the decrease is almost double than what is observed in non-obese patients, considering only the patients with at least three months of follow-up (see below). Of note, diet choice is not significantly different for the obese and non-obese patients. 

### 4.4. Special Populations: Diabetic Patients

No demographic or kidney-function difference is observed between diabetic (95% type-2 diabetes), and non-diabetic patients, except for the Charlson Index and BMI which are, as expected, higher in the diabetic population (Table 5). As for the Charlson Index, the differences reflect a higher average comorbidity in diabetic patients (complicated diabetes counts for 2 units in this index), but extreme comorbidity (Charlson Index of 10 or above) does not significantly differ in the two subsets.

Diet choice is affected by the fact that more patients in the non-diabetic group had contraindications to protein restriction (neoplasia, short life expectancy, malnutrition); the differences are offset if these patients are not considered. No difference in dietary compliance is observed. Of note, glycated hemoglobin remained stable (with a non significant trend towards improvement), underlining the importance of a comprehensive dietary management (Table 5).

### 4.5. Nutritional Parameters and Compliance

Table 6 reports the main biochemical and compliance data in patients with at least 3 months of follow-up and with biochemical data available in the same CHM laboratory. Adherence to the diets prescribed was considered good in about three fourths of the population, without significant differences according to type of diet (normalization 72%; moderately restricted: traditional diet 76%; supplemented plant based diet 75%).

A mean reduction of 0.2–0.4 g/kg/day was observed in all subsets, and was highest in patients for whom normalization of protein intake was a priority (−0.4 g/kg/day). In this short period of time, kidney function remained overall stable. Serum albumin increased in all subsets by 0.1–0.2 g/dL (significance was reached for the overall cohort and for normalization and traditional diets, suggesting that the nutritional intervention was at least not detrimental in the short term). 

The median reduction of protein intake in obese patients was higher than in non-obese patients (obese patients: 0.38 g/kg/day (CI: 0.32–0.40), versus non obese patients: 0.22 g/kg/day (CI: 0.22–0.32)). No difference was observed between diabetic and non-diabetic patients (median reduction 0.32 (CI: 0.20–0.46) versus 0.36 (CI: 0.31–0.36) g/kg/day).

### 4.6. Characteristics of Patients Who Started Dialysis

Seven patients started dialysis; the main clinical features are reported in Table 7. Dialysis was unplanned in two cases, both affected by a cardio-renal syndrome, in the context of an acute cardiac decompensation. The remaining patients started dialysis with an incremental policy (1–2 sessions per week); three patients started dialysis, discontinued it and started it again during the follow-up period. 

### 4.7. Logistic Regression: Compliance

Logistic regression analysis was performed on the subset of 65 patients with at least 3 months of follow-up and available data in the same laboratory (Table 6).

Two different outcomes were tested: in the first the outcome was “good compliance” with the diet prescribed, regardless of the target intake. In the second, the outcome was attainment of a target intake of 0.6 g/kg/day (“good diet”). Different models were used to account for the partial collinearity between the Charlson Index and two clinically relevant covariates (age and diabetes); in one model only obesity was analyzed, to avoid collinearity with diabetes (Table 8 and Table 9). 

The only variable that showed a significant correlation with outcome was obesity, which is associated with a higher probability of reaching the prescribed goals. This holds true both in the case of adherence to the prescribed diet and in the case of reaching the 0.6 g/kg/day goal.

## 5. Discussion

Adherence to dietary management is a critical issue in CKD patients. In spite of the potential benefit of low-protein diets, some authors hold that their intrusiveness in daily life, and the difficulty in obtaining adherence offset the advantages and fail to make these diets an interesting option for most patients [38,39].

The present paper takes issue with this negative outlook, and the main result of our study is to suggest that good compliance can be obtained from the majority of patients, including elderly subjects, diabetics and obese patients, categories that are usually considered to be particularly reluctant to change their dietary habits (Table 6).

The setting of the study is probably a relevant element: the patients described were followed up in a recently opened unit dedicated to the care of patients with advanced CKD in a large hospital in a rural area of France. The Unit, called UIRAV (acronym for Unité pour l’Insuffisance Rénale AVancée, Unit for the Care of Advanced CKD) offers integrated care for patients with stages 4–5 CKD, who are not on dialysis, and for selected patients with progressive stage-3 CKD. The caregiver team is stable, consisting of two nephrologists, three dietitians, one psychologist and a small nursing group. In this setting, all patients undergo dietary evaluation, and progressive reduction of protein intake is a goal for all those that do not present contraindications, including malnutrition or very short life expectancy, in which the goal is usually avoiding hypercatabolism and stabilizing clinical condition. 

Only 10 patients in our unit (7.6%) were not enrolled in a program of protein restriction (Table 2); in fact the importance of comorbid conditions is also shown by the fact that they represent 6 of the 9 patients who died during this period, for reasons closely linked to the same conditions that contraindicated protein restriction (Table 3). 

The UIRAV adapted to the French setting a choice-based approach to diet, which was previously developed in Italy, in a similar unit (Table 1, Figure 1 and Figure 2) [33,34]. The need to adapt the diet program to an older population with higher comorbidity, and a different background of dietary habits, and without readily available protein-free food, led us to modify the diet options, with a wider use of “traditional diets” with mixed proteins, frequently integrated with alpha-ketoacid and amino acid supplements, to compensate for proteinuria or to try to mediate between the need to stabilize kidney function and attention to maintaining good nutritional status, which is particularly precarious in the elderly (Table 2). 

One of the main results of our study is to show that, while a consistent reduction in protein intake, ranging from 0.2 to 0.4 g/kg/day, was observed in all diet subsets, and about 75% of the patients were considered to have attained the diet goals, albumin levels increased in all subsets; the overall increase is statistically significant (*p* = 0.001; Table 6).

The reasons of this almost paradox increase, that needs confirmation on a larger scare, are probably a combination of factors including attention to the overall nutritional care (controlling adequate caloric intake before reducing protein intake, correcting eventual errors in usual diets, favoring “high quality” food, reducing snacks and sweets and also paying attention to the quality of lipids), and the use of supplementation by ketoacids and amino acids in the case of relevant proteinuria or borderline-low serum albumin levels. In any case, the increase is not related to weight loss, or to hemoconcentration (as also witnessed by the stable hemoglobin levels).

While in Italy, according the previous experience on which the current system is inspired, patients were usually enrolled in a program of low protein diets since enrollment; in France, the approach we developed to protein reduction was stepwise (Figure 1 and Figure 2) and started from normalization of the protein intake. In fact, in the setting of study, as almost everywhere in France, the protein intake is well above the 0.8 g/kg/day, which is presently indicated as the goal for the overall population [40,41]. Interestingly, all subsets of patients, regardless of the diet they were following, attained the goal of a reduction of protein intake of at least 0.2 g/kg/day, an important, and often neglected “relative goal”, according to a series of studies that suggest that, independently from the final protein intake, each decrease of at least 0.2 g/kg/day of protein is associated with a reduction in the deterioration of kidney function [29,30,31]. 

While this short-term implementation study is not aimed at assessing the effect of the prescribed diets on kidney function, nor their potential in retarding or avoiding dialysis, eGFR did not significantly decrease over this short period of observation.

Furthermore, it is worth noting that with the exclusion of the two patients in the study affected by a cardio-renal syndrome, who presented frequent episodes of decompensation and started unplanned dialysis, incremental dialysis start, increasingly considered as the best therapeutic option, was possible for all other patients (Table 7) [10,11,42].

Focusing attention on particular populations, our cohort is evenly divided between diabetic and non-diabetic patients, while about 40% of the patients are obese (Table 4 and Table 5). Choice of diet and compliance with protein restriction was not significantly different in diabetic and non-diabetic patients, nor was the overall reduction in protein intake in these two subsets (Table 5). These results are in line with previous experience with the same dietary approach in an Italian setting, and allow us to confirm that, at least in elderly, type-2 diabetic patients, who were the overwhelming majority of our patients, diabetes per se is neither a contraindication for this type of dietary management, nor is it associated with poor therapeutic compliance [43]. 

One of the most interesting results of our study regards obese patients, a population that is overrepresented in the French cohort, where it accounts for about 40% of the subjects (Table 4). In spite of the fact that these patients are usually considered resistant to any type of dietary changes, obese patients displayed an impressive reduction in protein intake (almost 0.4 g per kg of real body weight), and overall good compliance (Table 4).

This finding was confirmed in the multivariate analysis, that identifies in obesity an element significantly associated with compliance, differently from sex, age, comorbidity, diabetes and eGFR (Table 8 and Table 9). The reasons why obese patients were more prone than non-obese patients to reduce their protein intake are not fully clear, and will be the object of a future more detailed analysis, focused in detail on dietary habits. On one side, it has to be acknowledged that it is much easier to reduce a very high intake than a lower one, and that obese patients usually have a larger “margin of maneuver” in reducing protein intake, also on the account of the fact that the calculation was based upon the actual body weight, except for the cases with extreme obesity. However, the reduction is remarkable (almost double with respect to non-obese patients) and we may speculate that this may also be the reflection of the “cultural” acknowledgement of the importance of reducing body weight, linked to the educational campaigns for weight reduction in France. In any case, this quite unexpected finding underlines the importance of including obese CKD patients in dietary follow-up, so they can benefit from the same methods for retarding dialysis, which are better established in non-obese patients.

Our study has several limitations, mainly because of its short follow-up, and limited number of patients; since it is a feasibility study, and focused on compliance, it could be seen as a pilot experiment in further extending the use of low-protein diets in settings in which experience is still limited and the baseline diet differs from the Mediterranean and Asian diets, which are often considered easier to adapt to protein restriction.

The strength of the study is therefore in its suggestions for clinical implementation and further research.

## 6. Conclusions

Reducing protein intake is also feasible in a population with high comorbidity, including the presence of about 50% of diabetic patients, and in elderly individuals. A personalized approach, with a differentiated offer may be the key to success, understood as good dietary adherence, which was observed, in our study, in about 75% of the cases, regardless of age, eGFR, diabetes and comorbidity, thus stressing the importance of offering this treatment option to all patients in whom severe contraindications are not found.

While long-term follow-up in a larger patient population is needed to confirm these promising results on a larger scale, two data findings merit particular attention. The first one is the finding of a significant increase in serum albumin, in spite of a significant reduction in protein intake, which stresses that reducing protein intake is not necessarily associated with malnutrition, and may instead suggest that increasing protein intake is not an effective way to contrast hypoalbuminemia.

The second one is that obese patients can show very good adherence to protein restriction, thus suggesting that these patients, often considered as being refractory to changes in diet, should not be forgotten as they may benefit from the same dietary measures traditionally offered to non-obese patients. 

## Figures and Tables

**Figure 1 nutrients-11-00036-f001:**
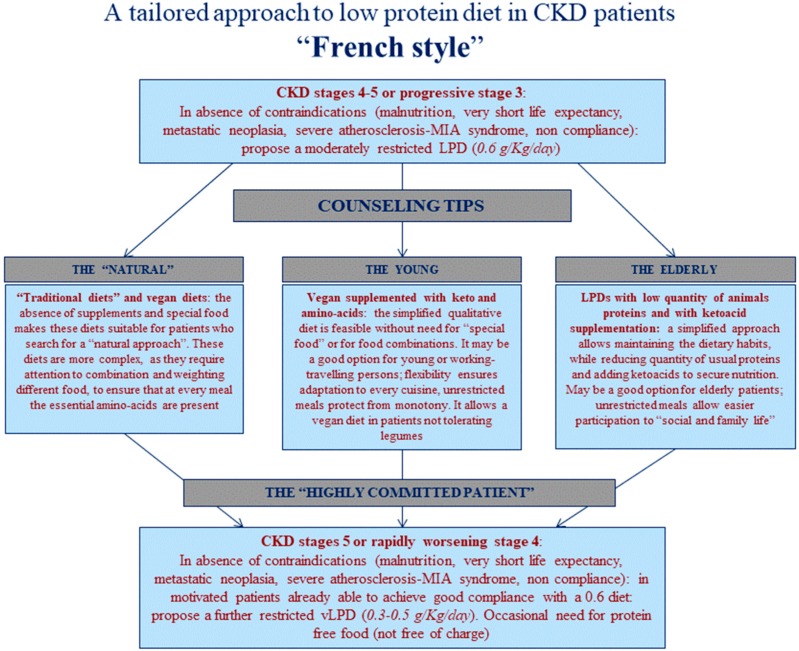
The multiple-choice “French style” model. CKD: Chronic Kidney Disease, MIA: Malnutrition Inflammation Atherosclerosis, LPS: Low Protein Diet.

**Figure 2 nutrients-11-00036-f002:**
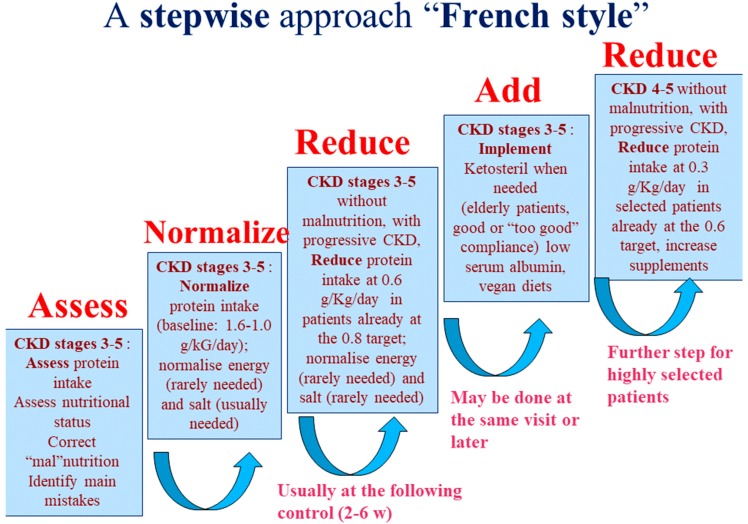
The stepwise approach. CKD: Chronic Kidney Disease.

**Figure 3 nutrients-11-00036-f003:**
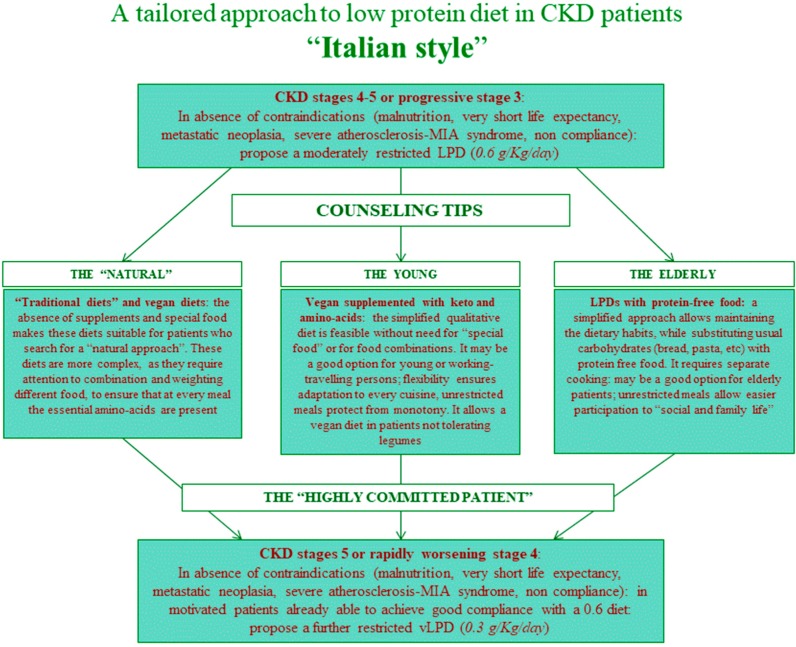
Multiple-choice reference “Italian style” model. CKD: Chronic Kidney Disease, MIA: Malnutrition Inflammation Atherosclerosis, LPS: Low Protein Diet.

**Figure 4 nutrients-11-00036-f004:**
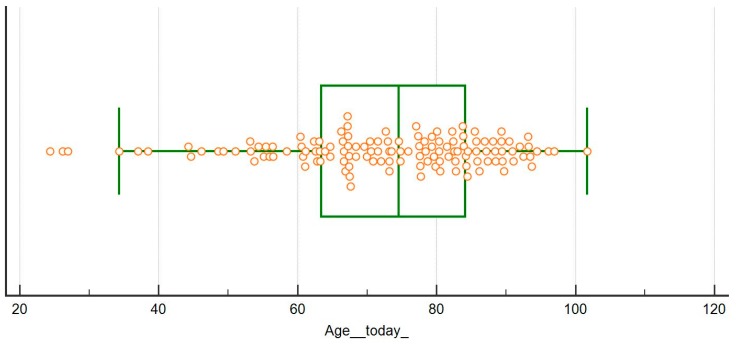
Age at study in the French population.

**Figure 5 nutrients-11-00036-f005:**
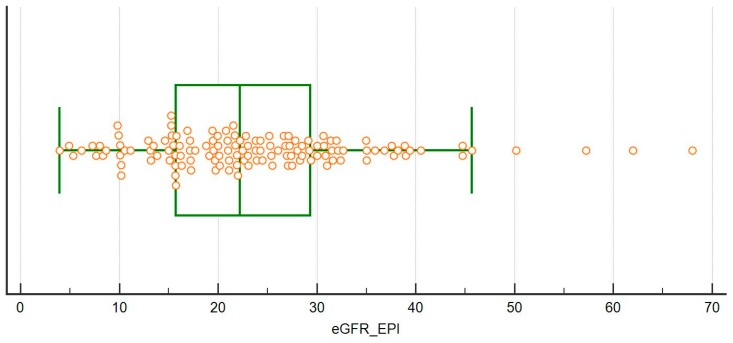
Distribution of e-GFR (CKD-EPI) at recruitment.

**Table 1 nutrients-11-00036-t001:** Baseline characteristics of the study group, compared with the Italian reference group.

First Diet	French Cohort (UIRAV)	Italian Reference Cohort	*p*
N	131	457	
Males, *n* (%) Females, *n* (%)	82 (62.6%) 49 (37.4%)	281 (68.5%) 176 (38.5%)	0.818 *
Age (years), median (min-max)	74 (24–101)	70 (19–97)	0.124
Age over 65, *n* (%)	95 (72.6%)	274 (60%)	**0.010 ***
Age over 80, *n* (%)	49 (37.4%)	73 (15.4%)	**<0.001 ***
CCI, median (min-max)	8 (2–14)	7 (2–13)	**0.018**
CCI ≥ 7, *n* (%)	93 (71%)	248 (54.8%)	**0.001 ***
CCI ≥ 10, *n* (%)	22 (16.8%)	71 (15.5%)	0.728 *
Diabetes, *n* (%)	66 (50.4%)	150 (32.8%)	**<0.001 ***
Cardiopathy, *n* (%)	30 (22.9%)	210 (46%)	**<0.001 ***
Neoplasia, *n* (%)	17 (13%)	99 (21.7%)	**0.028 ***
BMI (kg/m^2^), median (min-max)	28.3 (16.7–51.2)	26.1 (13.3–51.4)	0.115
sCreatinine (mg/dL), median (min-max)	2.6 (1.3–10.4)	2.8 (0.5–74)	0.225
eGFR-EPI (mL/min), median (min-max)	22 (4–68)	20 (3–127)	0.302
GFR < 15 (mL/min) at enrollment, *n* (%)	24 (18.3%)	125 (27.4%)	**0.036 ***
GFR < 10 (mL/min) at enrollment, *n* (%)	11 (8.4%)	44 (9.8%)	0.670 *
Proteinuria (g/day), median (min-max)	0.5 (0.1–8.3)	0.80 (0.1–12)	0.09
Proteinuria ≥ 1 g/day, *n* (%)	50 (38.2%)	203 (44.4%)	0.203 *
Proteinuria ≥ 3 g/day, *n* (%)	22 (16.8%)	79 (17.3%)	0.895 *
Glomerulonephritis-systemic disease, *n* (%)	3 (2.3%)	95 (21.2%)	**<0.001 ***

Legend: CCI: Charlson’s Comorbidity Index. E-GFR EPI: GFR according to the CKD-EPI equation. ADPKD: autosomal dominant polycystic kidney disease. “*” indicates that the Chi-Square Independence Test was used, otherwise, the Kruskal-Wallis Test was used. Bold *p* values have significant differences with alpha error at 5%.

**Table 2 nutrients-11-00036-t002:** Baseline characteristics of the population, on the basis of diet choice.

First Diet	Normalization	Moderate Restriction Traditional	Moderate Restriction Plant-Based Supplemented	No Protein Restriction	*p* between Groups
N	75	24	22	10	
Males, *n* (%) Females, *n* (%)	47 (62.7%) 28 (37.3%)	18 (75%) 6 (25%)	11 (50%) 11 (50%)	6 (60%) 4 (40%)	0.377
Age (years), median (min-max)	78 (24–101)	74 (44–91)	70 (34–89)	67 (44–88)	0.293
Age over 65, *n* (%)	54 (72%)	19 (79.2%)	15 (68.2%)	7 (70%)	0.853 *****
Age over 80, *n* (%)	36 (48%)	6 (25%)	4 (18.2%)	3 (30%)	0.031 *
CCI, median (min-max)	8 (2–12)	8 (4–14)	6.5 (2–12)	9 (5–12)	0.046
CCI ≥ 7, *n* (%)	55 (73.3%)	19 (79.2%)	11 (50%)	8 (80%)	0.104 *
CCI ≥ 10, *n* (%)	15 (20%)	1 (4.2%)	2 (9.2%)	4 (40%)	0.155 *
SGA A, *n* (%)	65 (86.7%)	23 (95.9%)	19 (86.4%)	4 (40%)	**<0.001 ***
SGA B, *n* (%)	10 (13.3%)	1 (4.2%)	2 (9.1%)	4 (40%)	
SGA C, *n* (%)	0 (0%)	0 (0%)	1 (4.6%)	2 (20%)	
MIS, median (min max)	5 (1–14)	5 (2–9)	5 (2–16)	9.5 (5–18)	**<0.001**
MIS ≥ 9, *n* (%)	8 (10.7%)	2 (8.3%)	5 (22%)	7 (70%)	**<0.001 ***
MIS ≥ 14, *n* (%)	1 (1.3%)	0 (0%)	1 (4.6%)	2 (20%)	**0.009** *
Diabetes, *n* (%)	37 (49.3%)	17 (70.8%)	10 (45.5%)	2 (20%)	0.047 *
Cardiopathy, *n* (%)	23 (30.7%)	1 (4.2%)	5 (22.7%)	1 (10%)	0.041 *
Neoplasia, *n* (%)	4 (5.3%)	3 (12.5%)	3 (13.6%)	7 (70%)	**<0.001 ***
sCreatinine (mg/dL), median (min-max)	2.3 (1.3–7.6)	2.7 (1.5–9.6)	3.3 (1.7–7.7)	2.6 (1.3–10.4)	**0.004**
eGFR-EPI (mL/min), median (min-max)	24 (8–68)	22 (5–40)	15 (5–46)	23 (4–50)	**0.012**
eGFR < 15 (mL/min) at enrollment, *n* (%)	7 (9.3%)	3 (12.5%)	12 (54.5%)	2 (20%)	**<0.001 ***
GFR < 10 (mL/min) at enrollment, *n* (%)	3 (4%)	2 (8.3%)	5 (22.7%)	1 (10%)	0.050 *
Proteinuria (g/day), median (min-max)	0.3 (0.1–8.3)	1.2 (0.1–4.5)	1.8 (0.1–7)	0.4 (0.1–3)	**0.003**
Proteinuria ≥ 1 (g/day), *n* (%)	19 (25.3%)	14 (58.3%)	14 (63.6%)	3 (30%)	**0.001 ***
Proteinuria ≥ 3 (g/day), *n* (%)	8 (10.7%)	5 (20.8%)	8 (36.4%)	1 (10%)	0.034 *
Glomerulonephritis-systemic disease, *n* (%)	0	0	2 (9.1%)	1 (10%)	0.087 *
Ketoanalogues, *n* (%)	10 (13%)	0	22 (100%)	1 (10%)	**<0.001 ***

Legend: CCI: Charlson’s Comorbidity Index. E-GFR EPI: GFR according to the CKD-EPI equation. ADPKD: autosomal dominant polycystic kidney disease. “*” indicates that the Chi-Square Independence Test was used, otherwise, Kruskal-Wallis Test was used. Bonferroni Correction involves a *p* value at 0.0125 to adapt the test for multiple comparison; significant results have been indicated in bold.

**Table 3 nutrients-11-00036-t003:** Characteristics of the patients who died.

Sex	Age	CCI	MIS	SGA	Kidney Disease	CKD Stage	Cause of Death	Protein Restriction	Dialysis
M	67	12	11	B	NAS	4	Neoplasia (liver)	None (short life expectancy)	no
F	44	9	10	B	Interstitial nephropathy	4-5	Neoplasia (lung)	None (short life expectancy, PEW)	no
F	49	8	7	A	FSGS	5	Heart failure (primary pulmonary hypertension)	None (short life expectancy, non compliance)	PD
M	84	7	9	B	NAS	4	Cardiac death	Normalisation	no
M	88	8	10	A	NAS	4	Popliteal artery rupture	Normalisation	no
M	88	10	9	C	NAS	5	Cardiac death	None (short life expectancy, patient’s choice)	no
M	81	7	3	A	NAS	5	Cardiac death	Normalisation	no
F	60	6	14	C	Diabetic nephropathy	5	Hemorrhage due to (voluntary) section of the dialysis catheter	None (PEW, non compliance)	HD
M	65	11	18	C	Interstitial nephropathy	3B	Neoplasia (lung)	None (short life expectancy, PEW)	no

Legend: M: male, F: female; MIS: malnutrition inflammation score; SGA: subjective global assessment (A: well nourished, B: moderate malnutrition; C: severe malnutrition); CCI: Charlson Comorbidity Index; CKD: chronic kidney disease; PD: Peritoneal dialysis; HD: hemodialysis-hemodiafiltration; NAS: nephroangiosclerosis, PEW: protein energy wasting.

**Table 4 nutrients-11-00036-t004:** Particular populations: obese patients.

First Diet	BMI ≥ 30 kg/m^2^	BMI < 30 kg/m^2^	*p*
N	53	78	
Males, *n* (%) Females, *n* (%)	29 (54.7%) 24 (45.3%)	53 (68%) 25 (32%)	0.126 *
Age (years) median (min-max)	73 (24–93)	76 (26–101)	0.221
Age over 65, *n* (%)	35 (66%)	60 (76.9%)	0.172 *
Age over 80, *n* (%)	17 (32%)	32 (41%)	0.301 *
CCI, median (min-max)	8 (2–14)	8 (2–12)	0.341
CCI ≥ 7, *n* (%)	40 (75.5%)	53 (68%)	0.354 *
CCI ≥ 10, *n* (%)	9 (17%)	13 (16.7%)	0.963 *
Diabetes, *n* (%)	37 (69.9%)	29 (37.2%)	**<0.001 ***
Cardiopathy, *n* (%)	15 (28.3%)	15 (19.2%)	0.227 *
Neoplasia, *n* (%)	3 (5.7%)	14 (18%)	**0.041 ***
sCreatinine (mg/dL), median (min-max)	2.5 (1.3–9.6)	2.6 (1.3–10.4)	0.683
eGFR-EPI (mL/min), median (min-max)	22 (5–57)	23 (4–68)	0.957
Proteinuria (g/day), median (min-max)	0.8 (0.1–8.3)	0.4 (0.1–6.5)	0.090
Proteinuria ≥ 1 (g/day), *n* (%)	24 (45.3%)	26 (33.3%)	0.169 *
Proteinuria ≥ 3 (g/day), *n* (%)	12 (22.7%)	10 (12.8%)	0.142 *
SGA A, *n* (%)	49 (92.5%)	62 (79.5%)	0.096 *
SGA B, *n* (%)	4 (7.6%)	13 (16.7%)	
SGA C, *n* (%)	0 (0%)	3 (3.9%)	
MIS, median (min-max)	5 (1–11)	5 (1–18)	0.974
MIS ≥ 9, *n* (%)	6 (11.3%)	16 (20.5%)	0.168 *
MIS ≥ 14, *n* (%)	0 (0%)	4 (5.1%)	0.095 *
Normalization of protein intake, *n* (%)	33 (62.3%)	42 (53.9%)	0.120 *
Moderate restriction traditional, *n* (%)	12 (22.6%)	12 (15.4%)	
Plant-based supplemented, (0.6) *n* (%)	7 (13.2%)	15 (19.2%)	
No restriction, *n* (%)	1 (1.9%)	9 (11.5%)	
BMI (kg/m^2^) median (min-max)	34.8 (51.2–30.3)	25.9 (29.8–16.7)	**<0.001**
BMI ≥ 35, *n* (%)	25 (47.2%)	0 (0%)	**<0.001 ***
BMI ≥ 40, *n* (%)	10 (18.9%)	0 (0.00%)	**<0.001 ***

Legend: M: male, F: female; MIS: malnutrition inflammation score; SGA: subjective global assessment (A: well nourished, B: moderate malnutrition; C: severe malnutrition); CCI: Charlson Comorbidity Index; CKD: chronic kidney disease. “*” indicates that the Chi–Squared Independence Test was used, otherwise, the Kruskal-Wallis Test was used. Bold *p* values have significant differences with alpha error at 5%.

**Table 5 nutrients-11-00036-t005:** Particular populations: diabetic patients.

First Diet	Diabetes	No Diabetes	*p*
N	66	65	
Males, *n* (%) Females, *n* (%)	42 (63.6%) 24 (36.4%)	40 (61.5%) 25 (38.5%)	0.804 *
Age (years), median (min-max)	72 (26–101)	77 (24–96)	0.175
Age over 65, *n* (%)	46 (67.9%)	49 (75.4%)	0.468 *
Age over 80, *n* (%)	22 (33.3%)	27 (41.5%)	0.334 *
CCI, median (min-max)	8 (5–14)	7 (2–12)	**<0.001**
CCI ≥ 7, *n* (%)	57 (86.4%)	36 (55.4%)	**<0.001** *
CCI ≥ 10, *n* (%)	13 (19.7%)	9 (13.8%)	0.372 *
Cardiopathy, *n* (%)	19 (28.8%)	11 (16.9%)	0.108 *
Neoplasia, *n* (%)	3 (4.5%)	14 (21.5%)	**0.004** *
sCreatinine (mg/dL), median (min-max)	2.6 (1.3–7.7)	2.58 (1.3–10.4)	0.945
eGFR-EPI (mL/min), median (min-max)	22 (7–57)	23 (4–7)	0.872
Proteinuria (g/day), median (min-max)	0.7 (0.1–8.7)	0.4 (0.1–7)	0.364
Proteinuria ≥ 1 (g/day), *n* (%)	28 (42.4%)	22 (33.9%)	0.088 *
Proteinuria ≥ 3 (g/day), *n* (%)	11 (16.7%)	11 (16.9%)	0.969 *
SGA A, *n* (%)	59 (89.39%)	52 (80%)	
SGA B, *n* (%)	5 (7.6%)	12 (18.5%)	
SGA C, *n* (%)	2 (3%)	1 (1.5%)	0.161 *
MIS, median (min-max)	5 (1–17)	5 (1–18)	0.589
MIS ≥ 9, *n* (%)	8 (12.1%)	14 (21.5%)	0.151 *
MIS ≥ 14, *n* (%)	3 (4.6%)	1 (1.5%)	0.319 *
Normalization of protein intake, *n* (%)	37 (56.1%)	38 (58.5%)	**0.047**
Traditional (0.6), *n* (%)	17 (25.6%)	7 (10.8%)	
Plant-based supplemented (0.6), *n* (%)	10 (15.2%)	12 (18.5%)	
No restriction, *n* (%)	2 (3%)	8 (12.3%)	
BMI (kg/m^2^), median (min-max)	31 (16.7–51.2)	27 (18.3–50)	**0.001**
BMI ≥ 35, *n* (%)	17 (26%)	8 (12%)	0.051 *
BMI ≥ 40, *n* (%)	6 (9.1%)	4 (6.2%)	0.528 *
HbA1c at enrollment (Pre) (%), median (min-max) At last update (Post) (%), median (min-max)	Pre 7.1 (5.1–11.2) Post 6.7 (4.9–9.4)		0.104

Legend: M: male, F: female; MIS: malnutrition inflammation score; SGA: subjective global assessment (A: well nourished, B: moderate malnutrition; C: severe malnutrition); CCI: Charlson Comorbidity Index; CKD: chronic kidney disease. “*” indicates that the Chi–Squared Independence Test was used, otherwise, the Kruskal-Wallis Test was used. Bold *p* values have significant differences with alpha error at 5%.

**Table 6 nutrients-11-00036-t006:** Compliance and biochemical data realized in the same unit for patients with at least 3 months of on-diet follow-up.

	Normalization (0.8) N = 32			Traditional (0.6) N = 17			Plant Based Supplemented (0.6) N = 16			All Cases N = 65		
	Base Line	Last Update	*p*	Base Line	Last Update	*p*	Base Line	Last Update	*p*	Base Line	Last Update	*p*
“Good adherence” (%)		72%			76%			75%			74%	0.934 ≠
Protein intake, g/kg/day median (min-max)	1.2 (0.7–1.5)	0.8 (0.6–1.4)	**<0.001** *	0.9 (0.6–1.3)	0.7 (0.5–0.9)	**<0.001** *	1.1 (0.5–1.5)	0.7 (0.4–1)	**<0.001** *	1.1 (0.5–1.5)	0.7 (0.4–1.4)	**<0.001** *
Creatinine, mg/dL mean (± SD)	2.4 (± 0.6)	2.6 (± 0.8)	0.088	2.9 (± 1)	3.1 (± 1.7)	0.369	3.6 (± 1.5)	3.9 (± 1.8)	0.550	2.84 (± 1.10)	3.05 (± 1.46)	**0.013**
Proteinuria g/24 h median (min-max)	0.4 (0.1–6.2)	0.5 (0.1–6.2)	0.149 *	1.2 (0.1–4.5)	1.4 (0.1–2)	0.999 *	1.4 (0.1–5.7)	2.5 (0.1–5.8)	0.893 *	0.6 (0.1–6.16)	0.7 (0.1–6.16)	0.566 *
Proteinuria ≥ 1 g/24 h *n* (%)	9 (28%)	7 (22%)	0.625 †	10 (59%)	10 (59%)	0.999 †	9 (56%)	8 (50%)	0.999 †	28 (43%)	25 (38%)	0.453 †
Albumin, g/dL median (min-max)	**3.5** **(3.1–4)**	**3.6** **(2.9–4.2)**	**0.037 ***	**3.5** **(2.9–3.9)**	**3.7** **(2.9–4)**	**0.034 ***	**3.2** **(2.8–3.9)**	**3.3** **(2.7–3.9)**	**0.216 ***	**3.5** **(2.8–4)**	**3.6** **(2.7–4.2)**	**0.001 ***
Albumin < 3 g/dL *n* (%)	0 (0%)	1 (3%)	0.999 †	2 (12%)	0 (0%)	0.999 †	4 (25%)	2 (13%)	0.625 †	6 (9%)	4 (6%)	0.687 †
PTH, ng/L median (min-max)	92.5 (27–252)	84 (18–614)	0.423 *	115 (33–328)	105 (44–490)	0.720 *	145.5 (13–188)	113 (24–251)	0.229 *	101 (13–328)	91 (18–614)	0.941 *
BUN, mg/dL median (min-max)	47.5 (26–79)	46.4 (26.3–99.4)	0.135 *	48 (26–121)	43.1 (30.5–74.2)	0.329 *	57.5 (25–81)	57.1 (25–124.1)	0.900 *	48 (25–121)	47.1 (25–124.1)	0.811 *
HCO3 mmol/L median (min-max)	24.5 (18–34)	23 (16–33)	0.150 *	24 (16–34)	24 (16–32)	0.599 *	21 (17–31)	22 (17–31)	0.770 *	24 (16–34)	23 (16–33)	0.193 *
Hemoglobin, g/dL mean (± SD)	12.6 (± 1.4)	12.3 (± 1.4)	0.251	12.4 (± 1.5)	12.1 (± 1.1)	0.332	11.3 (± 1.6)	11.1 (± 1.1)	0.570	12.2 (± 1.6)	12.9 (± 1.3)	0.108
GFR CKD-EPI mL/min/1.73 m^2^ median (min-max)	23 (15–57)	24 (12–52)	0.140 *	22 (10–35)	21 (7–44)	0.854 *	15 (7–38)	14.5 (6–36)	0.110 *	22 (7–57)	21 (6–52)	0.053 *
GFR MDRD, mL/min/1.73 m^2^ median (min-max)	24 (15–55)	25 (12–50)	0.153 *	23 (10–34)	23 (8–45)	0.934 *	15 (7–42)	15.5 (6–32)	0.110 *	22 (7–57)	21 (6–50)	0.071 *

For quantitative data, “*” indicates non-parametric data were analyzed with the Wilcoxon Test, otherwise, data were analyzed with the T-Test. The McNemar Test was used for pre-post qualitative data, and is represented by “†“, for comparison between all groups- “≠” indicates that the Chi -Square Independence Test was used. Bold *p* values have significant differences.

**Table 7 nutrients-11-00036-t007:** Characteristics of the patients who started dialysis.

Sex	Age	CCI	MIS	SGA	Kidney Disease	Type of Dialysis	Diet	Days in UIRAV to Dialysis Start
M	69	10	7	A	Cardio renal syndrome	HD–urgent (cardio-renal)	Normalization	79 §
F	49	8	7	A	FSGS and cardio renal syndrome	PD incremental *	None	§
M	82	8	7	A	Cardio renal syndrome	HD–urgent (cardio-renal)	Normalization	88
M	79	7	6	A	NAS	HD incremental **	0.6 plant-based	26 §§
F	53	4	5	A	APL syndrome	HD incremental	0.6 traditional	§
F	77	8	8	A	Diabetic nephropathy	HD incremental	0.6 traditional	98
F	60	6	14	C	Diabetic nephropathy	HD incremental *	None (low irregular intake)	§

Legend: M: male, F: female; MIS: malnutrition inflammation score; SGA: subjective global assessment (A: well nourished, B moderate malnutrition; C severe malnutrition); CCI: Charlson Comorbidity Index; CKD: chronic kidney disease; PD: Peritoneal dialysis; HD hemodialysis-hemodiafiltration; NAS: nephroangiosclerosis. FSGS: focal segmental glomerulonephritis; APL: antiphospholipid syndrome; UIRAV: unit dedicated to advanced CKD. * patients who died; ** grafted patients; § followed up after first dialysis start, kidney function partially recovered, and dialysis was needed again after 1–6 months; §§ previously followed by the nephrology service.

**Table 8 nutrients-11-00036-t008:** Logistic regression analysis: outcome: “good compliance” according to the prescribed diet (patients with at least 3 months of follow-up).

Model including Charlson Comorbidity Index, Age				
	Crude OR (95% CIs)	*p*	Adjusted OR (95% CIs)	*p*
Sex: Female	0.98 (0.67–1.45)	0.951	0.60 (0.12–2.99)	0.535
CCI < 7	2.02 (0.88–4.64)	0.100	2.29 (0.47–1.16)	0.302
eGFR-EPI (mL/min) ≥ 20	1.07 (0.33–3.46)	0.908	1.13 (0.27–4.65)	0.863
Age (years) < 65	1.64 (0.46–5.82)	0.444	1.43 (0.28–7.32)	0.668
BMI (kg/m^2^) ≥ 30	**5.16 (1.29–20.57)**	**0.013**	**3.89 (0.89–16.79)**	**0.070**
**Model including Diabetes, Age**
Sex: Female	0.98 (0.67–1.45)	0.951	1.98 (0.45–8.77)	0.794
eGFR-EPI (mL/min) ≥ 20	1.07 (0.33–3.46)	0.908	1.32 (0.36–4.84)	0.675
Age (years) < 65	1.64 (0.46–5.82)	0.444	1.98 (0.45–8.78)	0.365
BMI (kg/m^2^) ≥ 30	**5.16 (1.29–20.57)**	**0.013**	**5.70 (1.32–24.73)**	**0.020**
Diabetes: no	1.30 (0.42–4.07)	0.652	0.89 (0.24–3.29)	0.868
**Model including Age only**
Sex: Female	0.98 (0.67–1.45)	0.951	0.83 (0.20–3-36)	0.791
eGFR-EPI (mL/min) ≥ 20	1.07 (0.33–3.46)	0.908	1.31 (0.36–4.76)	0.685
Age (years) < 65	1.64 (0.46–5.82)	0.444	2.04 (0.48–8.74)	0.337
BMI (kg/m^2^) ≥ 30	**5.16 (1.29–20.57)**	**0.013**	**5.52 (1.35–22.60)**	**0.017**

Legend: CCI: Charlson Comorbidity Index; BMI: body mass index; e-GFR-EPI: estimated glomerular filtration rate according to the CKD-EPI formula. Bold values represent significant data for *p* < 0.05.

**Table 9 nutrients-11-00036-t009:** Logistic regression analysis: outcome: “0.6 diet” followed with good compliance (patients with at least 3 months of follow-up).

Model including Charlson Comorbidity Index, Age				
	Crude OR (95% CIs)	*p*	Adjusted OR (95% CIs)	*p*
Sex: Female	0.67 (0.22–2.04)	0.479	1.19 (0.28–5.07)	0.749
Charlson Index < 7	0.67 (0.20–2.24)	0.511	0.43 (0.09–2.09)	0.295
eGFR-EPI (mL/min) ≥ 20	1.50 (0.53–4.26)	0.445	2.08 (0.55–7.86)	0.283
Age (years) < 65	0.83 (0.25–2.69)	0.750	0.87 (0.19–4.09)	0.865
BMI (kg/m^2^) ≥ 30	**4.29 (1.43–12.81)**	**0.007**	**5.97 (1.59–22.44)**	**0.008**
**Model including Diabetes, Age**				
Sex: Female	0.67 (0.22–2.04)	0.479	0.64 (0.18–2.24)	0.486
eGFR-EPI (mL/min) ≥ 20	1.50 (0.53–4.26)	0.445	1.62 (0.51–5.14)	0.410
Age (years) < 65	0.83 (0.25–2.69)	0.750	1.05 (0.27–4.09)	0.940
BMI (kg/m^2^) ≥ 30	**4.29 (1.43–12.81)**	**0.007**	**4.61 (1.42–14.97)**	**0.011**
Diabetes: no	1.61 (0.59–4.45)	0.353	1.02 (0.32–3.27)	0.971
**Model including Age only**				
Sex: Female	0.67 (0.22–2.04)	0.479	0.64 (0.18–2.24)	0.487
eGFR-EPI (mL/min) ≥ 20	1.50 (0.53–4.26)	0.445	1.63 (0.51–5.14)	0.408
Age (years) < 65	0.83 (0.25–2.69)	0.750	1.055 (0.28–3.94)	0.945
BMI (kg/m^2^) ≥ 30	**4.29 (1.43–12.81)**	**0.007**	**4.64 (1.51–14.29)**	**0.007**

Legend: CCI: Charlson Comorbidity Index; BMI: body mass index; e-GFR-EPI: estimated glomerular filtration rate according to the CKD-EPI formula. Bold values represent significant data for *p* < 0.05.

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
