# Peer review of "Moderate Protein Restriction in Advanced CKD: A Feasible Option in An Elderly, High-Comorbidity Population. A Stepwise Multiple-Choice System Approach"

_nutrients, 2018, doi:10.3390/nu11010036_

Reviewer 1 Report

Overview

The authors studied the feasibility of CKD patients (stages 3-4) successfully implementing protein restricted diets, given their reported benefit to delay the need for kidney replacement therapy and/or slow nephropathy progression.  They studied patients who were mostly > age 70 years, representing the largest age demographic of CKD patients, with a large (40%) proportion of patients with diabetes, again, realistically representing the spectrum of patients seen by nephrologists today.  Their "step-wise" approach to implementing diets of 0.8 g/kg bw/day, 0.6 g of "mixed" protein, or 0.6 g with plant-base protein supplemented with ketoacids led to compliance in 74% of patients.  Importantly, patients maintained adequate nutritional status, including a small but significant increase in serum albumin, over the 3 month follow up.

These studies are important given the challenge faced by nephrologists and dieticians to implement these protein restricted diets.  They are additionally important in that they were executed in the context of a society with much higher baseline protein intake than the highest 0.8 g restricted level studied.

Major Comments

The main novel finding reported appears to be that the author's stepwise approach, not just the "individualized" approach described in the conclusion of the abstract, was successful in the majority of patients studied.  Other studies have previously reported that protein restriction can be done without compromising nutritional status (Aparicio, et al.  JASN 11:708, 2000) so the authors showing this is not among the novel findings of this study.  This reviewer suggests that the authors emphasize this aspect of their study.

That being said, the fact that the authors report a net increase in serum albumin, at all 3 levels of prescribed protein restriction, is a very important finding.  As the authors state, this finding shows that dietary approaches to increase serum albumin must not necessarily include the inordinately high protein intakes that characterize the baseline intakes of many modern developed societies.  Can the authors speculate as to the mechanism(s) as to how dietary protein restriction increased serum albumin?

The authors report that they sought to reduce dietary sodium to < 6 g/day which is higher than that recommended by KDIGO and by the Cardiologists.  This would suggest that the benefits they describe can be achieved with a dietary sodium intake that patients in modern developed societies are more likely to tolerate.

The authors report that the patient assigned to the 0.6 g groups had slightly higher (0.7 g) protein intakes than prescribed.  Can the authors report if the excess protein was of the same prescribed to the patients or was the excess "off diet" protein?

Can the authors detail what was the proportion of animal-source and plant-source protein in the "mixed" protein dietary prescription?  Relatedly, what were the predominant foods in the "plant-base" prescription?

When the patients began their "step-wise" progression toward their assigned diet, did the follow up begin at the start of the "step-wise" intervention or did the follow up begin only after the patients achieved the prescribed level of restricted dietary protein?

For the patients with diabetes, can the authors comment on their glycemic control before and after protein restriction?

It is interesting that obesity was associated with increased compliance.  Can the authors speculate as to an explanation?

Minor Comments

"Enrolment" is misspelled.

Author Response

Reviewer 1.

The authors studied the feasibility of CKD patients (stages 3-4) successfully implementing protein restricted diets, given their reported benefit to delay the need for kidney replacement therapy and/or slow nephropathy progression.  They studied patients who were mostly > age 70 years, representing the largest age demographic of CKD patients, with a large (40%) proportion of patients with diabetes, again, realistically representing the spectrum of patients seen by nephrologists today.  Their "step-wise" approach to implementing diets of 0.8 g/kg bw/day, 0.6 g of "mixed" protein, or 0.6 g with plant-base protein supplemented with ketoacids led to compliance in 74% of patients.  Importantly, patients maintained adequate nutritional status, including a small but significant increase in serum albumin, over the 3 month follow up.

These studies are important given the challenge faced by nephrologists and dieticians to implement these protein restricted diets.  They are additionally important in that they were executed in the context of a society with much higher baseline protein intake than the highest 0.8 g restricted level studied.

Answer

Thanks for your kind remarks.

Major Comments

1. The main novel finding reported appears to be that the author's stepwise approach, not just the "individualized" approach described in the conclusion of the abstract, was successful in the majority of patients studied.  Other studies have previously reported that protein restriction can be done without compromising nutritional status (Aparicio, et al.  JASN 11:708, 2000) so the authors showing this is not among the novel findings of this study.  This reviewer suggests that the authors emphasize this aspect of their study.

Answer:

Thank you for this remark; indeed, the stepwise approach is the main novelty introduced to adapt the Italian favorable experience to the French system; while follow-up is still to short to be able to analyze this issue in detail, this approach clarifies why we analyzed in this context the normalization of protein intake.

Acknowledging this important remark, we stressed this point in the title (in green) an in the text, in the discussion and in the results.

Indeed, the follow-up period is too short to allow us analyzing the effect of the progression through the different steps of protein reduction, but this will be the next goal for analysis, once we will have gathered at least 2 years of follow-up of the Unit.

2. That being said, the fact that the authors report a net increase in serum albumin, at all 3 levels of prescribed protein restriction, is a very important finding.  As the authors state, this finding shows that dietary approaches to increase serum albumin must not necessarily include the inordinately high protein intakes that characterize the baseline intakes of many modern developed societies.  Can the authors speculate as to the mechanism(s) as to how dietary protein restriction increased serum albumin?

Answer. The reviewer are right in pointing out that is one of the most important findings of our study; we did not want to stress it too much before having it confirmed on a longer follow-up; thanks for rising this point. We further commented it as follows:

The reasons of this almost paradox increase, that needs confirmation on a larger scare, are probably a combination of factors including attention to the overall nutritional care (controlling adequate caloric intake before reducing protein intake, correcting eventual errors in usual diets, favoring “high quality” food, reducing snacks and sweets and paying also attention to the quality of lipids), and the use of supplementation by ketoacid and aminoacids in the case of relevant proteinuria or borderline-low serum albumin levels.  In any case, the increase is not related to weight loss, or to hemoconcentration (as also witnessed by the stable hemoglobin levels).

3. The authors report that they sought to reduce dietary sodium to < 6 g/day which is higher than that recommended by KDIGO and by the Cardiologists.  This would suggest that the benefits they describe can be achieved with a dietary sodium intake that patients in modern developed societies are more likely to tolerate.

Answer

Thank you for this important remark; we further explained this issue as follows: the world health organisqtion sets the recommendation at 2 g sodium/day (equivalent to 5 g salt/day) (7). We added a mage of tolerance to 6 g of salt (NaCl). We therefore modified the text as follows:

Sodium intake is normalized where needed, with a first goal to less than 6 g of NaCl per day, which is slightly higher than the recommendation of the World health organization (2 g sodium/day, equivalent to 5 g salt/day), but is in line with a policy of progressive normalization of all crucial intakes, to favor compliance.  Due to the wide use of diuretics in our population, further adjustments are done, when needed, on the basis of the sodium level, of the overall diet and of hydration level.

4. The authors report that the patient assigned to the 0.6 g groups had slightly higher (0.7 g) protein intakes than prescribed.  Can the authors report if the excess protein was of the same prescribed to the patients or was the excess "off diet" protein?

Answer: 

The reason is probably linked to both; indeed, we considered compliance as good when protein intake was equal to the prescribed amount with a tolerance interval of 20% for 0.6 diets and of 15% for normalisation; this choice acknowledges the day to day variability and stresses the importance of overall reduction, also independently from reaching the exact target, in line with the stepwise approach previously discussed.

5. Can the authors detail what was the proportion of animal-source and plant-source protein in the "mixed" protein dietary prescription? Relatedly, what were the predominant foods in the "plant-base" prescription?

Answer: 

thanks for stressing this point: the following explanation was added to the summary information available in the figures.

“Traditional” diets are based upon the analysis of the usual dietary patterns in the area (for example, they rely on one mainly vegetarian meal per day, based upon a vegetable soup, containing potatoes as a source of starch, and small portions of dairy products, according to the usual habits in the French countryside), and maintain that about 50% of the protein intake should be of animal origin. Conversely plant-based diets rely on carbohydrates as potatoes, rice, bread, pasta as main sources of calories, and privilege proteins of vegetable origin (from grains and beans); since these protein sources are usually incomplete as for essential amminoacids, supplementation with a mixture of aminoacids and ketoacids (Kestosteril, available free of charge for CKD patients in France) is added. The dose, in keeping with the previous Italian experiences, is of 1 tablet per 8-10 Kg of body weight, to be further adjusted on the basis of albumin levels or protein losses [25, 33-34].

6. When the patients began their "step-wise" progression toward their assigned diet, did the follow up begin at the start of the "step-wise" intervention or did the follow up begin only after the patients achieved the prescribed level of restricted dietary protein?

Answer: 

patients are followed up during the whole process. This was better clarified in the methods as follows: The patients are followed-up by the nephrology and dietetics group throughout the whole process of protein reduction; moreover, the patients who do not undergo a protein restricted diet are also followed-up to timely identify and try to compensate for the eventual signs of malnutrition.

7. For the patients with diabetes, can the authors comment on their glycemic control before and after protein restriction

Answer:

Thanks for this remark: we added the data on the glycated hemoglobin to table 5: it remained stable (with even a trend towards improvement, in keeping with the results of the overall improved attention to a comprehensive dietary management.

We also added the following statement:

Of note, glycated hemoglobin remained stable (with a non significant trend towards improvement), underlining the importance of a comprehensive dietary management (table 5).

8. It is interesting that obesity was associated with increased compliance.  Can the authors speculate as to an explanation?

Answer

Thanks for this comment: we summarized our ideas in the following lines:

The reasons why obese patients were more prone than non obese patients to reduce their protein intake are not fully clear, and will e the object of a future more detailed analysis, focused in detail on dietary habits. On one side, it has to be acknowledged that it is much easier to reduce a very high intake than a lower one, and that obese patients usually have a larger “marge of maneuver” in reducing protein intake, also on the account of the fact that the calculation was based upon the actual body weight, except for the cases with extreme obesity. However, the reduction is remarkable (almost double with respect to non obese patients) and we may speculate that this may also be the reflection of the “cultural” acknowledgement of the importance of reducing body weight, linked to the educational campaigns for weight reduction in France. In any case, this quite unexpected finding underlines the importance of including obese CKD patients in dietary follow-up, so they can benefit from the same methods for retarding dialysis, which are better established in non-obese patients.

Minor Comments

"Enrolment" is misspelled. Ok, thanks this was corrected.

With the hope that the present version may have answered to the main points and questions, the authors would like to thank the reviewer for the keen comments and the efforts made to improve the quality of our report.

Reviewer 2 Report

I read with interest the article entitled “Moderate protein restriction in advanced CKD: a feasible option in an elderly, high-comorbidity population. A multiple-choice system approach” submitted for publication in Nutrients. This manuscript is relevant and appropriate to this journal as it narrates the application of an established protein restriction protocol in Italy to a population in France. Additionally, the applicability of very low, low, and moderate protein restriction diets in CKD, and particularly the eldery, is limited.  I hope the authors will find the following comments helpful in revising their manuscript:

Affiliations/credentials

·         Please modify dietician to dietitian.

Abstract

·         For the results section, the standard deviation (SD), range, or 95% CI is needed in addition to the mean/median.

·         I would recommend adding the percentage next to the number of patients that picked normalization, moderately protein-restricted diets, and plant-based diets.

·         For the 75 patients that chose normalization, please ad characteristics (age, CCI, eGFR), just as in the other groups.

Introduction

·         Page (P) 22, line (L) 61: define “extremely high”

·         P2 L69: in Italy or France, do doctors decide the nutritional treatment?

·         P2 L71-76: I would recommend splitting this sentence, it is long. Maybe period before “Likewise”?

Methods

·         Why was the KDOQI staging was chosen instead of the KDIGO grades? The authors had access to urinary protein as well?

·         P3 L113-114: what signs of PEW?

·        Was the prescription of protein for overweight and obese patients based on actual weight, ideal body weight, or adjusted body weight?

·         Any references for the Italy model?

·         For the figures, please add a legend with the abbreviations.

·         In Figure 1, does the VLPD is supplemented with ketoacids?

·         For energy intake, what recommendation was given to patients with obesity?

·         P5 L149, what is a day hospital? Outpatient clinic?

·         P6 L182: why is “The” red?

·         Was statistically significance set as α ≤0.05?

·         When was protein normalization or restriction started? From the Table 1 GFR ranges, there were participants with GFR of 68ml/min.

·         Besides protein and energy, it would be very interesting to assess dietary intake and dietary patterns change with the adoption of normalized-protein or restricted protein diets.

Results

·         For the tables, please modify the P values that are marked as 0.000 to<0.001.

·         P10 L246: please modify protein malnutrition to protein-energy wasting. 

The manuscript is supposed to assess the feasibility of the restriction of protein in the elderly. Why were there patients as young as 24y? Which were also considered for the analysis of “special populations”, such as those patients with obesity and diabetes mellitus.

·         P13 L292: from table 6, not all subsets improved in albumin. Those that were on the moderate reduction plant-based supplemented was not statistically significant.

Discussion

·         I do not believe that obese patients are excluded from studies that may offer slow progression of CKD. Do the authors have any references for this? I do agree that there may be an implicit bias in some healthcare professionals on how some patients with obesity may resist change, but this is should not be a generalization. I would be interested to see the point of view of the psychologist in your team.

Author Response

 Reviewer 2.

Comments and Suggestions for Authors

I read with interest the article entitled “Moderate protein restriction in advanced CKD: a feasible option in an elderly, high-comorbidity population. A multiple-choice system approach” submitted for publication in Nutrients. This manuscript is relevant and appropriate to this journal as it narrates the application of an established protein restriction protocol in Italy to a population in France. Additionally, the applicability of very low, low, and moderate protein restriction diets in CKD, and particularly the eldery, is limited.  I hope the authors will find the following comments helpful in revising their manuscript:

Thanks for your kind words; we really appreciated your interesting comments,

Affiliations/credentials

·         Please modify dietician to dietitian.

Answer, Ok, thanks, done.

Abstract

·         For the results section, the standard deviation (SD), range, or 95% CI is needed in addition to the mean/median.

·         I would recommend adding the percentage next to the number of patients that picked normalization, moderately protein-restricted diets, and plant-based diets.

·         For the 75 patients that chose normalization, please ad characteristics (age, CCI, eGFR), just as in the other groups.

Answer: thanks we added the information (in yellow)

Introduction

·         Page (P) 22, line (L) 61: define “extremely high”

answer: thanks: we added the following precision: Mortality is extremely high, especially in the first months after dialysis start, where, according to the DOPPS data, it may reach up to 33 deaths per 100 patient-years, and this finding should induce us to focus on two main issues: how to safely prolong the dialysis-free interval, and how to smooth the transition to dialysis [9-11].

·         P2 L69: in Italy or France, do doctors decide the nutritional treatment?

Answer: yes, it is in both countries diet is a medical prescription.

·         P2 L71-76: I would recommend splitting this sentence, it is long. Maybe period before “Likewise”?

Answer: thanks, done.

Methods

·         Why was the KDOQI staging was chosen instead of the KDIGO grades? The authors had access to urinary protein as well?

Answer: yes, we have access to proteinuria, but we chose the stages since the unit is dedicated to patients with advanced reduction of the kidney function, and the level of proteinuria is taken into consideration for the dietary choice and eventually for adding ketoacid and aminoacid supplementation.

·         P3 L113-114: what signs of PEW?

Answer, thanks for making this point: the following sentence was added:

All patients are followed up to identify signs of protein energy wasting (PEW), such as reduction in body weight (unexplained by edema reduction), reduction in lean body mass (clinical assessment, and integration with bioimpedance on demand), reduction in serum albumin, prealbumin or total proteins, especially in the absence of acute inflammatory event, or other clinical marker of poor nutrition, from the dietary journal, assessed by the dietitians, to vitamin deficits or unexplained anemia.

·        Was the prescription of protein for overweight and obese patients based on actual weight, ideal body weight, or adjusted body weight?

Thanks for the precision; It was based on actual weight up to a BMI of 35 Kg/m2 and to adjusted body weight above this limit. The comment was added.

In the case of obese patients, the prescription of protein was based on actual weight up to a BMI of 35 Kg/m2 and to adjusted body weight above this limit.

·         Any references for the Italy model?

Yes, they are given in several points, and further added in the methods: 25, 33-34].

·         For the figures, please add a legend with the abbreviations.

Thanks, this was completed.

·         In Figure 1, does the VLPD is supplemented with ketoacids?

All vLPDs are supplemented with ketoacids and aminoacids, this was better underlined as follows in the methods:

Very low protein diets are always integrated by ketoacids and aminoacids (Ketosteril, 1 tablet per 5 Kg of real body weight).

·         For energy intake, what recommendation was given to patients with obesity?

Thanks again for pointing to this important detail: we added the following precisionin the methods:

Energy intake is established in a case-by-case basis in obese patients, on the account not only of caloric intake, but also of the daily activities, privileging, wherever possible, increasing physical activity to reducing energy below 25 Kcal of adjusted weight per day. 

·         P5 L149, what is a day hospital? Outpatient clinic?

Yes, it corresponds to a “one-day hospitalization”, where all tests and consultations are free of charge.

We added this precision in the methods:
(Outpatient clinic dedicated to treatments that have to be done in-hospital, but do not need overnight stay, such as iron infusion, or iv diuretics, or to complex diagnostic assessments, including at least three consultations or imageries).

·         P6 L182: why is “The” red?

By mistake, it was corrected.

·         Was statistically significance set as α ≤0.05?

yes, thanks, this was added.

·         When was protein normalization or restriction started? From the Table 1 GFR ranges, there were participants with GFR of 68ml/min.

The reviewer is right: GFR was above 45 in 4 cases, who presented a rapid reduction in the kidney function, and either a single kidney or nephrotic proteinuria. The distribution of e-GFR is as follows; the figure and the following sentence were added:

The distribution of the kidney function at recruitment (CKD-EPI) is depicted in figure 5; of note eGFR was above 45 ml/min in 4 cases, who were started on protein restriction on the account of a combination of rapid decrease of the kidney function, nephrotic proteinuria or single kidney.

·         Besides protein and energy, it would be very interesting to assess dietary intake and dietary patterns change with the adoption of normalized-protein or restricted protein diets.

Answer: thanks for the suggestion; indeed we hope to be able to get into further details in a next paper dedicated to obese patients.

Results

·         For the tables, please modify the values that are marked as 0.000 to<0.001.< span="">

thanks, this was done.

·         P10 L246: please modify protein malnutrition to protein-energy wasting.

The manuscript is supposed to assess the feasibility of the restriction of protein in the elderly. Why were there patients as young as 24y? Which were also considered for the analysis of “special populations”, such as those patients with obesity and diabetes mellitus.

Answer: the population is overall an elderly one; this does not mean that we did not treat younger patients; we added a figure, and stressed that this is a population with high median age.

 P13 L292: from table 6, not all subsets improved in albumin. Those that were on the moderate reduction plant-based supplemented was not statistically significant.

Answer: All subsets improved albumin levels, but, as correctly pointed out, the increase is not statistically significant in all of them, mainly for the small number of cases in some subsets; this was clarified in the text as follows:

Serum albumin increased in all subsets by 0.1-0.2 g/dl (significance was reached for the overall cohort and for normalization and traditional diets, suggesting that the nutritional intervention was at least not detrimental in the short term).

Discussion

·         I do not believe that obese patients are excluded from studies that may offer slow progression of CKD. Do the authors have any references for this? I do agree that there may be an implicit bias in some healthcare professionals on how some patients with obesity may resist change, but this is should not be a generalization. I would be interested to see the point of view of the psychologist in your team.

Answer: You are right; diabetic patients are often excluded, and obese patients are not always clearly described… We tried to clarify it better in rephrasing the discussion.

As for the psychologist in our team… well we do not have one specifically aimed at following-up on diet patients, we hope to be able to recruit a person in particular for the follow-up of obese CKD patents.
With the hope that the present version may have answered to the main points and questions, the authors would like to thank the reviewer for the keen comments and the efforts made to improve the quality of our report. 

Reviewer 3 Report

Present study by Fois A et al beneficial effect of protein restriction in elderly patients with respect to nephropathy progression in place of renal replacement therapy. Patients were given normalized protein (0.8 g/kg), moderate protein (0.6 g/kg) , and traditional 0.6 g/kg)  protein mixed with plant based diet.  Patients are about 70 years old and half of them are diabetic individuals. It really interesting study comparing effect of protein intake in CKD patients. It definitely good piece of information of “Nutrients” journal audience. Here are the comments

1.     Current study explain about protein restriction in CKD patients but half of the patients had diabetes and obesity. What is the status of glycemic control? That might directly worsen the CKD.

2.     Both dietary approach and the outcome of this study is similar to Italian settings. Other than confirming in French CKD population, what is the novelty of this study? Even reference cohort has more power than this study. Indeed, it is hard to differentiate current study with previous one.

3.     What guidelines author followed to restrict the sodium amount to less than 6 g. Because AHA recommends 2.3g/day but an ideal limit of no more than 1,500 mg per day.

4.     How and why does CKD stage 4&5 patients are not undergone dialysis?

5.     In what context the current approach is “minimalist outlook”?

6.     Discussion: Break and rewrite the sentence starting with …The Unit, ….to…nursing group.

7.     Discussion: Replace retarded with …potential in reducing the use of dialysis.

Author Response

Reviewer 3

Present study by Fois A et al beneficial effect of protein restriction in elderly patients with respect to nephropathy progression in place of renal replacement therapy. Patients were given normalized protein (0.8 g/kg), moderate protein (0.6 g/kg) , and traditional 0.6 g/kg)  protein mixed with plant based diet.  Patients are about 70 years old and half of them are diabetic individuals. It really interesting study comparing effect of protein intake in CKD patients. It definitely good piece of information of “Nutrients” journal audience. Here are the comments

1.       Current study explain about protein restriction in CKD patients but half of the patients had diabetes and obesity. What is the status of glycemic control? That might directly worsen the CKD.

Answer: Thanks for the important comment: the glycemic control was actually non-significantly improved, and we added the Hb1ac results in the table 5

Of note, glycated hemoglobin remained stable (with a non significant trend towards improvement), underlining the importance of a comprehensive dietary management (table 5).

2.       Both dietary approach and the outcome of this study is similar to Italian settings. Other than confirming in French CKD population, what is the novelty of this study? Even reference cohort has more power than this study. Indeed, it is hard to differentiate current study with previous one.

Answer: Well, this is actually the point, as the demonstration that the same system that works in Italy, a Mediterranean Country, in which the baseline habits are more easily adapted to the reduction in protein intake; this was better stressed as follows in the text:

Introduction:

the present study tests the hypothesis that personalization of nutritional approach represents a feasible option for first of all normalizing and then reducing protein intake in a CKD population with high median age and comorbidity, followed up in Le Mans, France, a region where baseline protein intake is high and dietary habits are more meat-based than plant-based.

Methods :

This multiple-choice stepwise approach represents an adaptation to French dietary patterns of a “diet system” previously set up in Italy, a country in which it was possible to rely on widespread acceptance of a Mediterranean baseline pattern, and reduction of protein intake is facilitated by the availability of protein-free commercial food, provided free of charge to CKD patients (figure 3 )

Discussion:

 The need to adapt the diet program to an older population with higher comorbidity, and a different background of dietary habits, and without readily available protein-free food, led us to modify the diet options, with a wider use of “traditional diets” with mixed proteins, frequently integrated with alpha-ketoacid and aminoacid supplements, to compensate for proteinuria or to try to mediate between the need to stabilize kidney function and attention to maintaining good nutritional status, which is particularly precarious in the elderly (table 2).

3.       What guidelines author followed to restrict the sodium amount to less than 6 g. Because AHA recommends 2.3g/day but an ideal limit of no more than 1,500 mg per day.

Answer: when talking about 6 grams, we were mentioning “salt” and not sodium; Indeed the indication of the WHO and of most societies is to reduce NaCl (salt) below 5 grams; we remained a little higher to improve overall compliance, and not to make too many changes at the same time. The issue was better clarified as follows:

Sodium intake is normalized where needed, with a first goal to less than 6 g of NaCl per day, which is slightly higher than the recommendation of the World health organization (2 g sodium/day, equivalent to 5 g salt/day), but is in line with a policy of progressive normalization of all crucial intakes, to favor compliance.  Due to the wide use of diuretics in our population, further adjustments are done, when needed, on the basis of the sodium level, of the overall diet and of hydration level.

4.       How and why does CKD stage 4&5 patients are not undergone dialysis?

Dialysis is not indicated in stage 4 CKD and, in stage 5, we followed the presently advides intent to defer policy. This was clarified as follows:

Dialysis start is decided in CKD stage 5 on the basis of the clinical picture (assessment includes anorexia, weight loss, nausea, malnutrition, restless leg syndrome, poorly controlled hypertension); GFR, urea levels, water and acid-base balance, calcium-phosphate-PTH balance, anemia and serum albumin are taken into account in the evaluation of the patient.

5.       In what context the current approach is “minimalist outlook”?

We rephrased as: The present paper takes issue with this negative outlook

6.     Discussion: Break and rewrite the sentence starting with …The Unit, ….to…nursing group.Ok, thanks

7.     Discussion: Replace retarded with …potential in reducing the use of dialysis.

We rephrased the sentence as follows:

While this short-term implementation study is not aimed at assessing the effect of the prescribed diets on kidney function, nor their potential in retarding or avoiding dialysis, eGFR did not significantly decrease over this short period of observation.
With the hope that the present version may have answered to the main points and questions, the authors would like to thank the reviewer for the interesting and important comments and the efforts made to improve the quality of our report.

Round  2

Reviewer 2 Report

Thank you to the authors for addressing my comments/suggestions. At this time, I do not have further comments.

Reviewer 3 Report

I am pleased with revised version of this manuscript.